# Spectrum of BRCA1/2 Mutations in Romanian Breast and Ovarian Cancer Patients

**DOI:** 10.3390/ijerph19074314

**Published:** 2022-04-04

**Authors:** Radu Vidra, Tudor Eliade Ciuleanu, Adina Nemeș, Oana Pascu, Ana Maria Heroiu, Nicoleta Antone, Andreea Iulia Vidrean, Cristina Marinela Oprean, Laura Ancuta Pop, Ioana Berindan-Neagoe, Rares Eniu, Alexandru Eniu

**Affiliations:** 1Oncology Department, “Iuliu Hatieganu” University of Medicine and Pharmacy, 400012 Cluj-Napoca, Romania; radu.vidra@irgh.ro (R.V.); tudor_ciuleanu@hotmail.com (T.E.C.); adina.nemes@umfcluj.ro (A.N.); 2The Oncology Institute “Ion Chiricuta”, Republicii Street, No 34-36, 400015 Cluj-Napoca, Romania; pascu.oanateodora@gmail.com (O.P.); ani_vlazan@yahoo.com (A.M.H.); nicoleta.antone@iocn.ro (N.A.); 3Regional Institute of Gastroenterology and Hepatology “Octavian Fodor”, 400000 Cluj-Napoca, Romania; 4Medisprof Cancer Center, 400641 Cluj-Napoca, Romania; andreeavidrean@medisprof.ro; 5Department of Oncology, ONCOHELP Hospital, 200239 Timisoara, Romania; 6ANAPATMOL Research Center, “Victor Babes” University of Medicine and Pharmacy, 300041 Timisoara, Romania; 7Department of Oncology, ONCOMED Outpatient Unit Timisoara, 300239 Timisoara, Romania; 8Research Center for Functional Genomics, Biomedicine and Translational Medicine, “Iuliu Hatieganu” University of Medicine and Pharmacy, 400012 Cluj-Napoca, Romania; laura.ancuta.pop@gmail.com (L.A.P.); ioana.neagoe@umfcluj.ro (I.B.-N.); 9Faculte de Medecine, Universite de Geneve, 1206 Geneva, Switzerland; eniura@cardiff.ac.uk; 10Hopital Riviera-Chablais, 1847 Rennaz, Switzerland; aleniu@iocn.ro

**Keywords:** breast cancer, ovarian cancer, BRCA mutations, BRCA1 mutations, BRCA2 mutations

## Abstract

Background: About 10,000 women are diagnosed with breast cancer and about 2000 women are diagnosed with ovarian cancer each year in Romania. There is an insufficient number of genetic studies in the Romanian population to identify patients at high risk of inherited breast and ovarian cancer. Methods: We evaluated 250 women of Romanian ethnicity with BC and 240 women of Romanian ethnicity with ovarian cancer for the presence of damaging germline mutations in breast cancer genes 1 and 2 (BRCA1 and BRCA2, respectively) using Next-Generation Sequencing (NGS) technology. Results: Of the 250 breast cancer patients, 47 carried a disease-predisposing BRCA mutation (30 patients (63.83%) with a BRCA1 mutation and 17 patients (36.17%) with a BRCA2 mutation). Of the 240 ovarian cancer patients, 60 carried a BRCA mutation (43 patients (72%) with a BRCA1 mutation and 17 patients (28%) with a BRCA2 mutation). In the BRCA1 gene, we identified 18 variants (4 in both patient groups (ovarian and breast cancer patients), 1 mutation variant in the BC patient group, and 13 mutation variants in the ovarian cancer patient group). In the BRCA2 gene, we identified 17 variants (1 variant in both ovarian and breast cancer patients, 6 distinct variants in BC patients, and 10 distinct variants in ovarian cancer patients). The prevailing mutation variants identified were c.3607C>T (BRCA1) (18 cases) followed by c.5266dupC (BRCA1) (17 cases) and c.9371A>T (BRCA2) (12 cases). The most prevalent mutation, BRCA1 c.3607C>T, which is less common in the Romanian population, was mainly associated with triple-negative BC and ovarian serous adenocarcinoma. Conclusion: The results of our analysis may help to establish specific variants of BRCA mutations in the Romanian population and identify individuals at high risk of hereditary breast and ovarian cancer syndrome by genetic testing.

## 1. Introduction

According to Globocan, breast cancer (BC) in women is the most common cancer in the world, both in terms of incidence and mortality, and the incidence of BC is estimated to increase to approximately 33.8% of newly diagnosed cases by 2040. The same trend can be observed in ovarian cancer (OC), with an estimated increase in new cases of 36.6% by 2040, despite the advances in diagnostic techniques and the treatment of breast and ovarian cancer [1,2]. The multifactorial etiology of breast and ovarian cancer is responsible for this dire situation. Both environmental and genetic factors are included. The lifetime risk of breast and ovarian cancer is 45–80% in BRCA1 and BRCA2 mutation carriers [3,4,5]. BRCA1 and BRCA2 germline mutations are associated with an aggressive BC course (the triple-negative breast cancer subtype especially) and advanced OC [6,7]. According to a recent study, pathogenic mutations in the BRCA1 and BRCA2 genes confer a high risk of BC, contralateral BC, and OC. The average cumulative risk for BRCA1 carriers was estimated to be 60% for BC, 59% for OC, and 83% for contralateral BC. For BRCA2 carriers, the cumulative risk was estimated to be 55% for BC, 16.5% for OC, and 62% for contralateral BC [8,9].

The prevalence of BRCA1 and BRCA2 mutations varies between ethnic groups and geographical areas [10,11]. At present, the up-to-date BRCA Exchange database contains descriptions of more than 2100 pathogenic mutation variants in the BRCA1 gene and more than 2600 pathogenic mutation variants in the BRCA2 gene [12].

The spectrum of BRCA1 and BRCA2 gene alterations includes frameshift, nonsense, and missense mutations altering protein functions, splice mutations leading to truncation, and large rearrangements [13,14]. Founder mutations have been described in certain geographical areas by ethnic communities, such as Ashkenazi Jews and Icelandic, Dutch, Turkish, and Polish populations [15,16]. In Romania, BC is the most common female malignancy with an annual incidence of about 10,000 new cases and 3300 deaths [1,17]. In addition, about 2000 women are diagnosed with OC each year and about 1100 women with OC die of this disease each year [1]. To date, only a few surveys on BRCA1/2 mutations in Romanian patients with breast and ovarian cancer have been published [18,19]. The lack of a sufficient number of genetic studies in the Romanian population prevents us from developing an effective mutation screening tool that could identify patients at high risk of inherited breast and ovarian cancer.

Our goal in the present study was to identify the BRCA1/2 germline mutations in a cohort of 250 Romanian patients with BC and the BRCA1/2 germline/somatic mutations in 240 women with OC that satisfy the recognized international criteria for testing. This is one of the first comprehensive studies to determine the contribution of BRCA1/2 germline mutations to BC and OC development in the Romanian population.

## 2. Materials and Methods

### 2.1. Patients

Our study is a retrospective analysis of 250 patients with BC and 240 patients with OC diagnosed and treated at The Oncology Institute “Prof. Dr. Ion Chiricuta” Cluj-Napoca, Romania (OICN) from 2014 to 2019. All patients were of Romanian ethnicity. The clinical information (age, tumor grade and stage, luminal status, and surgical treatment), pathology reports, and family history were obtained from the patient/hospital registry. All patients were tested following the recognized Breast Cancer Linkage Consortium (BCLC) and National Comprehensive Cancer Network (NCCN) criteria (breast cancer before 40 years of age, triple-negative breast cancer before the age of 50, or having a conventional family history of breast cancer).

### 2.2. Genetic Analysis

#### 2.2.1. BC Cohort

##### Next-Generation Sequencing

The screening of BRCA genes for mutations was performed using genomic DNA from blood samples using the Purelink Genomic DNA kit (Invitrogen, Thermo Fisher Scientific, Waltham, MA, USA). 

The DNA was then evaluated using a NanoDrop spectrophotometer and concentrations between 10 and 158 ng/µL with 260/280 ratios of 1.5–2 were obtained. The quantified DNA was further used for sequencing library preparation using the Ion Ampliseq Community BRCA1_BRCA2 primer kit consisting of 167 amplicons to analyze the coding regions of both BRCA1/2 genes. For each sample, 10 ng/µL of DNA was used. To obtain the library’s amplicons, the following protocol was used. In the first PCR, we added 1 µL of AmpliSeq HiFi Master Mix, 3.5 µL of the primers of one of the three pools from the Ion Ampliseq Community BRCA1_BRCA2 kit, 1 µL of 10 ng/µL DNA, and 1.5 µL of nuclease-free water. The same amounts were used for each of the three pools of primers, so each sample had three PCR reactions. The program used for the PCR was: 1 cycle for 2 min at 99 °C, 20 cycles for 1 s at 99 °C and 4 min at 60 °C, and 1 cycle at 10 °C with an indefinite hold. After this PCR, the three reactions for each sample were put together in a new tube, 2 µL of FuPa reagent was added, and the mixture was incubated for 10 min at 50 °C, 10 min at 55 °C, 20 min at 60°, and a maximum of 60 min at 10 °C. In the next step, we added 4 µL of Swich solution, 2 µL of 1:4 diluted Ion Express Barcodes, and 2 µL of Ligase. The mixture was further incubated for 30 min at 22 °C, 5 min at 65 °C, 5 min at 72 °C, and a maximum of 60 min at 10 °C. After this step, the libraries were purified using Agencourt AmPure Beads (Beckman Coulter) and eluted in 40 µL of Platinum Polymerase and 1.6 µL of primers. Then, the mixture was incubated in a PCR with the following program: 2 min at 98 °C, 5 cycles at 98 °C for 15 sec and 64 °C for 1 min, and 1 cycle at 10 °C with an indefinite hold. After this PCR step, a two-step purification of the PCR products was performed using Agencourt AmPure Beads.

The libraries were quantified using Qubit 2.0 and the Qubit HS DNA Kit. All libraries were diluted to a concentration of 100 pM. Template preparation was performed using the Ion One Touch 200 OT2 Template kit (Applied Biosystems) with the Ion One Touch 2 System (Life Technologies). After the emulsion PCR, the libraries were enriched using the same kit as for the template preparation, together with the Ion One Touch ES apparatus. The enriched libraries were sequenced on 316 chips (8 libraries/chip) using the Ion PGM 200 sequencing kit and the Ion Torrent PGM (Life Technologies). After sequencing, the data were analyzed using the Torrent Suite and the Ion Reporter 5.0 software. The obtained mutations were verified in the ENIGMA, BIG, and ClinVar databases in order to identify the pathogenicity class for each mutation.

For the validation of the first 40 samples, the DNA from the first 40 samples was sent to Radboud University, Nijmegen, the Netherlands, where it was analyzed using the MiSeq platform (Illumina) according to the protocol provided by the manufacturer. 

##### Multiplex Ligation—Dependent Probe Amplification

For the evaluation of the Copy Number Variation and other large chromosomal variations for the BRCA1 and BRCA2 genes, we used the SALSA MLPA probe mix P-002-D1 BRCA1 and SALSA MLPA probe mix P045-B3 BRCA2/CHEK2 reagents from MRC Holland. For the MLPA reactions, we used the standard protocol provided by the manufacturer and 150 ng of DNA for each sample. The fragment analysis was performed on an ABI-Prism 310 Capillary sequencer (Applied Biosystems), and the fragment analysis was performed using the free Coffalyser software provided by MRC Holland. 

##### Sanger Sequencing

All the Class 3 and 5 mutations identified in the studied samples were validated by Sanger Sequencing using specific primers/assays from Thermo Scientific for each identified mutation. The PCR for the amplification of specific target regions of the BRCA1 and BRCA2 genes was performed using PCR SuperMix (Invitrogen) and 30–60 ng of DNA. After the initial PCR, the mixture was purified with CleanSweep PCR Purification reagent (Applied Biosystems) and the amplicons were quantified using Qubit 2.0. In the BigDye Termination reaction, we used 20 ng of amplicons and the BigDye Terminator v3.1 Cycle Sequencing kit (Applied Biosystems) with its specific protocol. After this reaction, the mixture was purified using the BigDye XTerminator Purification Kit (Applied Biosystems) using the protocol provided by the manufacturer. After the purification reaction, 20 µL of the mixture was analyzed on the ABI-Prism 310 Capillary sequencer (Applied Biosystems). The data obtained were transformed into FASTA files using the Chromas software and the FASTA files were analyzed using the MEGA 7 and Genome Browser software.

#### 2.2.2. Genetic Analysis of the OC Cohort

##### Next-Generation Sequencing

For the ovarian cancer cohort, we used two types of biological samples (blood and formalin-fixed paraffin-embedded tissue). The genomic DNA from the formalin-fixed paraffin-embedded tissue was extracted using the Code 405 MagCore^®^ Genomic DNA FFPE One-Step Kit (RBC Bioscience) kit and the MagCore HF16 automate platform, and the genomic DNA from blood samples was extracted using the NucleoSpin Tissue kit (Macherey&Nagel). Next-Generation Sequencing library preparation was performed using the Accel-Amplicon™ BRCA1, BRCA2 panel (Swift Biosciences), which covers all the coding regions of the BRCA1 and BRCA2 genes, including exon–intron junctions. Sequencing was performed on the MiSeq Next-Generation Sequencing benchtop (Illumina). After sequencing, the data were analyzed using the BWM-MEM algorithm for alignment and the GATK algorithm for variance characterization using the ClinVar, OMIM, 1000GENOMES, GnomeAD, Varsome, LOVD, BIC, and INVITAE databases using the NSClinical software. For missense mutations, the mutations were also analyzed in silico using the “dbNSFP” program.

The MLPA Analysis Was Performed Using the SALSA MLPA Probe mix P002 BRCA1 kit (MRC Holland) and the ABI3130xl Genetic Analyzer

The obtained data were analyzed using the Coffalyser.Net (MRC Holland) software.

### 2.3. Clinical Data Interpretation

The mutations identified were classified (into 5 classes) by consulting the ClinVar, BIC, and ENIGMA consortium databases. All patients included in this study previously signed a general institutional informed consent form for genetic testing. The study design was evaluated and approved by the Ethics Committee of OICN and the “Iuliu Hatieganu” (the University of Medicine and Pharmacy Cluj-Napoca Ethics Committee). Descriptive statistics were obtained in order to group subjects by categories such as age, histopathology subtype, tumor grade, stage of the disease, and surgical intervention. In addition, BC patients were categorized by hormone receptor status, HER2 status, unilateral BC, and bilateral BC.

## 3. Results

In this study, 250 Romanian BC patients and 240 Romanian patients with relapsed, high-grade, platinum-sensitive OC selected according to the established criteria were screened for mutations in the BRCA1/2 genes by direct sequencing.

### 3.1. Age and Other Clinical and Pathological Characteristics

In the BC group, the median age at diagnosis was 42 years (28–62 years old). A total of 57% of the patients (20 patients) were less than 40 years of age, 40% (14 patients) were between 41 and 60 years of age, and 3% (1 patient) were between 61 and 70 years of age.In the OC group, the median age at diagnosis was 50 years (32–83 years old). A total of 7% of the patients were less than 40 years of age, 81% were between 41 and 60 years of age, and 12% were between 61 and 83 years of age.The clinical and pathological characteristics of the BC and OC patients are summarized in Table 1 and Table 2.

### 3.2. BRCA Mutations in 47 BC and 60 OC Patients

In the BC patient group, following NGS, we identified 47 (18.8%) BRCA mutation bearers. A total of 63.83% of these patients (30 patients) had BRCA1 mutations and 36.17% (17 patients) had BRCA2 mutations.All the VUS and pathogenic mutations in the BC patient cohort were validated using Sanger sequencing.The MLPA analysis of BC patients revealed that all the included patients had normal copy number variations, except for one patient that exhibited a CHEK2-9(10) deletion but no pathogenic mutations in BRCA1 and BRCA2 genes. This deletion is considered to be a founder mutation in the Polish population.Among the OC mutation carriers, we identified 43 patients (72%) that had BRCA1 mutations and 17 patients (28%) that had the BRCA2 mutation. The clinical characteristics of all selected cases are summarized in Table 1 and Table 2.

#### 3.2.1. BRCA Mutation Variants

Regarding BRCA mutations in BC patients, we identified 8 mutations in the BRCA1 gene and 11 mutations in the BRCA2 gene. The mutation types identified were as follows: four frameshift deletions (21.05%) (two BRCA1 mutations and two BRCA2 mutations), one frameshift insertion (5.27%) (a BRCA1 mutation), six nonsense mutations (in a nonfunctional protein) (31.59%) (two BRCA1 mutations and four BRCA2 mutations), four missense mutations (nucleotide insertions) (21.05%) (one BRCA1 mutation and three BRCA2 mutations), and four splice site mutations (21.05%) (two BRCA1 mutations and two BRCA2 mutations) (Table 3).

#### 3.2.2. BRCA1 in the BC Population 

The BRCA1 mutational variants identified in the BC population that can be found in the HGVS database or the ClinVar database are the following.In the BRCA1 gene, we identified two frameshift deletions, one frameshift insertion, two nonsense mutations, one missense mutation, and two splice site mutations.The frameshift deletions c.4218delG (chr17:41234559) and c.68_69delAG (chr17:41276044), the frameshift insertion c.5266dupC (chr17:41209079), the two nonsense mutations c.3067C>T (chr17:41243941) and c.1687C>T (chr17:41245861), and the missense mutation c.181T>G (chr17:41258504) are BC/OC-causing mutations that have been previously reported in the literature (Table 4).The two splice site mutations c.4485-1G>T (chr17:41226539) and c.212+1G>T (chr17:41258472) are classified as pathogenic by the ClinVar database.The most prevalent mutation in the BC cohort was c.5266dupC (11 patients) (36.66%), followed by c.3067C>T (9 patients) (30%) and c.181T>G (4 patients) (13.33%).

#### 3.2.3. BRCA1 in the OC Population 

The BRCA1 mutational variants identified in the OC population that can be found in the HGVS database are the following. DNA sequencing revealed a new mutation, c.(80+1_81-1)_(441+1_442-1)del, which we could not find in the Breast Cancer Information Core (BIC) and Human Genome Variation Society (HGVS) databases (it is listed as VUS in the BIC database), with an unknown pathogenic effect and 16 deleterious mutations, of which 7 were frameshift mutations (c.135-2A>G, c.2411_2412delAG, c.3700_3704delGTAAA, c.4065_4068delTCAA, c.4675+1G>C, c5266dupC (5382insC), and c.843_846delCTCA), 4 were nonsense mutations (c.1066C>T, c.1687C>T, c.1789G>T, and c.3607C>T), 1 was a single nucleotide variant (c.4986+3G>C), and 4 were missense mutations (c.181T>G, c.5497G>A, c.558A>G, and c.556T>G) (Table 5).The most prevalent BRCA1 mutation was c.3607C>T 11 patients (26%), followed by c.5266dupC (5382insC) 8 patients (19%) and c.1687C>T and c.181T>G, each with an equal share of 9% (4 patients) (Table 5).

#### 3.2.4. BRCA2 in the BC Population

The BRCA2 mutational variants identified in the BC population that can be found in the HGVS database or the ClinVar database are the following. DNA sequencing revealed four nonsense mutations (c.4022C>G (chr13:32912514), c.1528G>T (chr13:32907143), c.8680C>T (chr13:32950854), and c.8695C>T (chr13:32950869)), three missense mutations (c.7007G>A (chr13:32921033), c.9371A>T (chr13:32968940), and c.8167G>C (chr13:32937506)), two frameshift deletions (c.9253delA (chr13:32954279) and c.5795_5796delAT (chr13:32914286)), and two splice site mutations (c.516+1G>A (chr13:32900420) and c.8755-1G>A (chr13:32953453)) (Table 6). 

The most prevalent mutation in the BC population was c.9371A>T (seven patients) (41.17%). All the other identified pathogenic mutations were observed in only one patient.

#### 3.2.5. BRCA2 in the OC Population

Regarding BRCA mutations in OC patients, we identified 17 BRCA1 mutations and 11 BRCA2 mutations. In the BRCA gene, we identified the following mutations: 25 frameshift mutations (44%) (17 BRCA1 mutations and 8 BRCA2 mutations), 18 nonsense mutations (31%) (17 BRCA1 mutations and 1 BRCA2 mutation), 13 missense mutations (23%) (6 BRCA1 mutations and 7 BRCA2 mutations), and 1 single nucleotide variant mutation (2%) (1 BRCA1 mutation) (Table 7).The BRCA2 mutational variants identified in the OC population that can be found in the HGVS database are the following. DNA sequencing revealed a new mutation, c.9658_9660delCCT, which has not been previously cited in the BIC and HGVS databases, with an unknown pathogenic effect (listed as VUS in the BIC database) and 10 deleterious mutations, of which 7 are frameshift mutations (c.1593delA, c.2435delA, c.3975_3978dupTGCT, c.5946delT, c.6267_6269delinsC, c.6839_6840insA, and c.8655dupA), 2 are missense mutations (c.9371A>T and c.3032C>G), and 1 is a nonsense mutation (c.3545_3546delTT) (Table 8).The most prevalent BRCA2 mutation in ovarian cancer patients was c.9371A>T (six patients) (35%), followed by c.6839_6840insA (two patients) (Table 9).

## 4. Discussion

This is one of the first comprehensive studies to evaluate the contribution of BRCA1/2 germline mutations to BC and OC development in the Romanian population.

Eight distinct BRCA1 mutational variations were identified in our BC patient group. Of these, four known variations have previously been reported to be pathogenic and one new mutation was identified that does not belong to the BIC and HGVS databases (it is listed as VUS in the BIC database). All the BRCA1 mutations identified in the BC cohort were identified as being pathogenic in the ClinVar database.

The most frequently encountered BRCA1 mutation in our BC group was chr17:41209079 (c.5266dupC). A total of 11 patients (36.37%) harbored this mutation, of which 6 had TNBC. Nine patients (30%) were found to have a c.3067C>T mutation, of which six had TNBC.

In the OC group, the situation was almost the same for the two most prevalent mutations, except that the prevailing BRCA1 mutation was c.3607C>T (11 patients) (26%) followed by c.5266dupC (5382insC) (8 patients) (19%).

Some BRCA mutational variations were found with a similar incidence in patient populations in some Central–Eastern European countries (Bulgaria, Turkey, Poland, the Czech Republic, Greece, and Belarus).

The third most prevalent BRCA1 mutation encountered in our BC patients, the missense variant c.181T>G, has been recognized as the second most frequent mutation in Poland and other European countries (Austria, the Czech Republic, and Belarus) [20]. In a recent study conducted by Dodova et al., the c.181T>G mutation was identified in Bulgarian breast cancer patients at a very low rate (two patients), and it was associated with early onset of the disease [20].

In our OC patients, the BRCA1 mutations c.181T>G and c.1687C>T were ranked fourth regarding prevalence. In our analysis, the c.181T>G mutation was found as a disease-causing mutation in four patients with TNBC and c.1687C>T was carried by two patients with TNBC and four patients with OC. These latter two mutations (c.181T>G and c.1687C>T) represent the main BRCA1 mutations in the Slovenian population with a frequency of 56% and 30%, respectively, and they are also found in South-Eastern Europe (Greece in particular) [21]. c.1687C>T also represents a common mutation prevalent in other European countries such as Austria and Sweden with a common founder ancestor [22].

Regarding the BRCA2 gene, we identified 11 different mutational variants in the BC patient cohort, all previously reported as being deleterious in the ClinVar database.

The most often encountered BRCA2 mutation in our BC group was the missense mutation chr13:32968940 (c.9371A>T) (seven patients (41.18%), of which only one had TNBC). This mutation has previously been identified in the Romanian population, but it is not very frequent in the populations around Romania [20]. It was also the most prevalent BRCA2 mutation among the OC patients that we analyzed.

The BRCA2 mutation c.8680C>T (chr13:32950854) has also been previously described in the Romanian population and, in our analysis, it was associated with early onset of BC (36 years) [22].

Another mutation that we found to be associated with the BRCA2 gene and is considered to be deleterious, c.7007G>A, is associated with abnormal splicing leading to premature translation and truncated proteins and was previously described in a Czech population [21].

We identified one new mutational variation, chr13:32912514 (c.4022C>G), that does not belong to the BIC and HGVS databases. c.4022C>G is implicated in FLNB-Related Spectrum Disorders, Joubert syndrome, Meckel–Gruber syndrome, “*Cutis laxa*” with severe pulmonary involvement, gastrointestinal and urinary abnormalities, and CC2D2A, PTCH1, LTBP4, and DMD-related disorders, but its significance is uncertain and its pathogenicity has conflicting interpretations. The BRCA2 mutational variant c.4022C>G has not been stated to be BC or OC pathogenic [23].

The identified BRCA mutational variants have no known association with an increased incidence of relapsed breast or ovarian cancer.

The mutation with the largest distribution across Eastern Europe is c.5263_5264insC [16]. In our analyzed groups of patients with breast and ovarian cancer and BRCA mutations, this mutation was not observed.

Mutations that have been frequently observed in neighboring populations (e.g., the Bulgarian and Polish populations) and mutational variants reported as being specific to a particular population include the BRCA1 mutations 185delAG (c.66_67delAG) and 5382insC (c.5263_5264insC) and the BRCA2 mutation 6174delT (c.5946delT) in Ashkenazi Jews [24], the BRCA1 mutations c.4035delA, C61G (c.-58C>G), and 5382insC (c.5263_5264insC) in a Polish population with familial breast cancer [25], the BRCA1 mutations c.303T>G, c.5324T>G, c.1623dupG, and c.4122_4123delTG in patients of African descent with familial breast cancer [26], the BRCA1 mutation Ex9-12del in Latin American, particularly Mexican, patients with familial breast and ovarian cancer [27], the BRCA2 mutation c.7480C>T in Korean patients with familial breast cancer [28], the BRCA1 mutation 5382insC (c.5263_5264insC) in Bulgarian patients with breast and ovarian cancer [20], and 5266dupC in the Trakya region of Turkey [29].

In conclusion, in our study, we identified 19 BRCA gene mutational variants, namely 8 BRCA1 variations and 11 BRCA2 variations, 4 of which were found in both groups of patients (ovarian and breast cancer patients). One particular mutation variant was found in the BC patient group and thirteen particular mutation variants were found in the OC patient group. We identified 17 BRCA2 variants (1 variant was discovered in both ovarian and breast cancer patients, 6 distinct variants were discovered in BC patients, and 10 distinct variants were discovered in OC patients).

The small number of BRCA1/2-positive BC and OC patients could be considered a limitation regarding the association between the BRCA1/2 mutational variations and the clinical features. This will be addressed in a further study, in which we will expand the patient cohort in order to obtain more significant clinical correlations and an increase in the statistical impact.

The prevailing mutation variants were identified to be c.3607C>T (BRCA1) (20 cases), c.5266dupC (BRCA1) (19 cases), and c.9371A>T (BRCA2) (13 cases) (Table 9).

## 5. Conclusions

The results of this study are useful because they identify specific variants of the BRCA mutations in Romanian patients with BC and OC. This study should be continued by investigating the practical implications of these results, namely by establishing correlations between the identified mutational subtypes, the clinical and pathological characteristics, and the evolutionary course of the neoplastic diseases BC and OC.

The second practical application of these results is in the identification of individuals at high risk of hereditary breast and ovarian cancer syndrome (HBOC) by genetic testing.

## Figures and Tables

**Table 1 ijerph-19-04314-t001:** Clinical and pathological characteristics of the 47 Romanian patients with primary BC and BRCA mutations selected by age at diagnosis and tumor characteristics.

Age (Years)	Number	Percentage
<40	26	55%
41–60	17	36%
>60	4	9%
**Bilateral BC**		
Yes	9	19%
No	38	81%
**TNBC**		
Yes	26	55%
No	18	38%
Unknown	3	7%
**Tumor hormone receptor status**		
ER		
Positive	19	40%
Negative	27	57%
Unknown	1	3%
**PR**		
Positive	16	34%
Negative	28	59%
Unknown	3	6%
**HER2 status**		
Positive	6	12%
Negative	36	76%
Unknown	5	12%
**Stage at diagnosis**		
I	3	6%
II	24	51%
III	11	23%
Unknown	9	19%
**Grade**		
2	11	23%
3	30	64%
Unknown	6	13%
Total	47 patients 100%	

**Table 2 ijerph-19-04314-t002:** Clinical and pathological characteristics of the 60 Romanian patients with primary OC and BRCA mutations selected by age at diagnosis and tumor characteristics.

	Number	Pertentage
**Age (years)**		
<40	4	7%
41–60	49	81%
>60	7	12%
**Histopathologic subtype**		
Serous adenocarcinoma	42	70%
Papilary serous adenocarcinoma	7	11%
Clear cell adenocarcinoma	1	2%
Papillary chistadenocarcinoma	5	8%
Serous chistadenocarcinoma	4	7%
Seromucinous chistadenocarcinoma	1	2%
**Stage at diagnosis**		
I	2	3%
II	5	9%
III	45	75%
IV	8	13%
**Grade**		
1	1	2%
2	3	5%
3	56	93%
**Surgical intervention**		
Yes	53	88%
No	7	12%
Total	60 patients	100%

**Table 3 ijerph-19-04314-t003:** BRCA1/2 mutations in the BC patient group.

Mutation	Number of Mutations (Number of Patients)	Percentage of Mutations (Percentage of Patients)	BRCA1 Number of Mutations (Number of Patients)	BRCA2 Number of Mutations (Number of Patients)
Frameshift deletion	4 (4)	21.05% (8.51%)	2 (2)	2 (2)
Frameshift insertion	1 (11)	5.27% (23.40%)	1 (11)	0
Nonsense	6 (15)	31.58% (31.91%)	2 (11)	4 (4)
Missense	4 (13)	21.05% (27.67%)	1 (4)	3 (9)
Splice site	4 (4)	21.05% (8.51%)	2 (2)	2 (2)

**Table 4 ijerph-19-04314-t004:** BRCA1 mutation rate (according to the Human Genome Variation Society (HGVS)) and its link to BC in the analyzed group.

	Locus	HGVS	Amino-Acid	Mutation	Nb.	Percentage
1	chr17:41209079	c.5266dupC	p.Gln1777fs	Frameshift Insertion	11	36.67%
2	chr17:41234559	c.4218delG	p.Lys1406fs	Frameshift Deletion	1	3.33%
3	chr17:41276044	c.68_69delAG	p.Glu23fs	Frameshift Deletion	1	3.33%
4	chr17:41243941	c.3607C>T	p.Arg1203Ter	Nonsense	9	30%
5	chr17:41245861	c.1687C>T	p.Gln563Ter	Nonsense	2	6.67%
6	chr17:41258504	c.181T>G	p.Cys61Gly	Missense	4	13.34%
7	chr17:41226539	c.4485-1G>T		Splice site	1	3.33%
8	chr17:41258472	c.212+1G>T		Splice site	1	3.33%
				Total	30	100%

**Table 5 ijerph-19-04314-t005:** BRCA1 mutations (according to the HGVS) and their links to OC in the analyzed group.

	HGVS	Cases	Percentage	Mutation Type
1	c.(80+1_81-1)_(441+1_442-1)del	1	2%	VUS
2	c.1066C>T	1	2%	Nonsense
3	c.135-2A>G	2	5%	Frameshift–splice acceptor
4	c.1687C>T	4	9%	Nonsense
5	c.1789G>T	1	2%	Nonsense
6	c.181T>G	4	9%	Missense
7	c.2411_2412delAG	1	2%	Frameshift deletion
8	c.3607C>T	11	26%	Nonsense
9	c.3700_3704delGTAAA (c.3695_3699GTAAA)	1	2%	Frameshift microsatellite
10	c.4065_4068delTCAA	1	2%	Frameshift deletion
11	c.4675+1G>C	1	2%	Frameshift–splice donor
12	c.4986+3G>C	1	2%	Single nucleotide variant
13	c.5266dupC (5382insC)	8	19%	Frameshift duplication
14	c.5497G>A	1	2%	Missense
15	c.5558A>G	1	2%	Missense
16	c.556T>G	1	2%	Missense
17	c.843_846delCTCA	3	7%	Frameshift deletion
	Total	43	100%	

**Table 6 ijerph-19-04314-t006:** BRCA2 mutations (according to the HGVS) and their links to BC in the analyzed group.

	Locus	HGVS	Amino-Acid ch.	Mutation	Nr.	Percentage
1	chr13:32907143	c.1528G>T	p.Glu510Ter	Nonsense	1	5.88%
2	chr13:32912514	c.4022C>G	p.Ser1341Ter	Nonsense	1	5.88%
3	chr13:32921033	c.7007G>A	p.Arg2336His	Missense	1	5.88%
4	chr13:32950854	c.8680C>T	p.Gln2894Ter	Nonsense	1	5.88%
5	chr13:32950869	c.8695C>T	p.Gln2899Ter	Nonsense	1	5.88%
6	chr13:32954279	c.9253delA	p.Thr3085Glnfs*19	Frameshift deletion	1	5.88%
7	chr13:32968940	c.9371A>T	p.Asn3124Ile	Missense	7	41.2%
8	chr13:32937506	c.8167G>C	p.Asp2723His	Missense	1	5.88%
9	chr13:32914286	c.5795_5796delAT	p.His1932fs	Frameshift deletion	1	5.88%
10	chr13:32900420	c.516+1G>A		Splice site	1	5.88%
11	chr13:32953453	c.8755-1G>A		Splice site	1	5.88%
				Total	17	100%

**Table 7 ijerph-19-04314-t007:** BRCA1/2 mutations in the OC group.

Mutation	Frequency	Percentage	BRCA1	BRCA2
Frameshift duplication	10	17%	8	2
Frameshift insertion	2	3%	0	2
Frameshift deletion	8	14%	5	3
Frameshift indel	1	2%	0	1
Frameshift splice donor	1	2%	1	0
Frameshift splice acceptor	2	3%	2	0
Frameshift microsatellite	1	2%	1	0
Nonsense deletion	1	2%	0	1
Nonsense	17	30%	17	0
Missense	13	23%	6	7
Single nucleotide variant	1	2%	1	0

**Table 8 ijerph-19-04314-t008:** BRCA2 mutations (according to the HGVS) and their links to OC in the analyzed group.

	HGVS	Cases	Percentage	Mutation Type
1	c.9658_9660delCCT	1	6%	VUS
2	c.1593delA	1	6%	Frameshift deletion
3	c.2435delA	1	6%	Frameshift deletion
4	c.3032C>G	1	6%	Missense
5	c.3545_3546delTT	1	6%	Nonsense deletion
6	c.3975_3978dupTGCT	1	6%	Frameshift duplication
7	c.5946delT	1	6%	Frameshift deletion
8	c.6267_6269delinsC	1	6%	Frameshift indel
9	c.6839_6840insA	2	11%	Frameshift insertion
10	c.8655dupA	1	6%	Frameshift duplication
11	c.9371A>T	6	35%	Missense
	Total	17	100%	

**Table 9 ijerph-19-04314-t009:** The prevailing mutation variants observed.

	Mutation	Breast	Ovary	Total
1	c.3607C>T (BRCA1)	9	11	20
2	c.5266dupC (BRCA1)	11	8	19
3	c.9371A>T (BRCA2)	7	6	13
4	c.181T>G (BRCA1)	4	4	8
5	c.1687C>T (BRCA1)	2	4	6

## Data Availability

The data that support the findings of this study are available from the corresponding author (C.M.O.), upon reasonable request.

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
