# Peer review of "Spectrum of BRCA1/2 Mutations in Romanian Breast and Ovarian Cancer Patients"

_ijerph, 2022, doi:10.3390/ijerph19074314_

Round 1

Reviewer 1 Report

Thank you for providing the opportunity to review the manuscript titled “Spectrum of BRCA 1/2 mutations in Romanian high risk breast and ovarian cancer patients” by Radu Vidra et.al. In this manuscript, Radu Vidra and co-authors investigates the mutational landscape of BRCA1 and BRCA2 in Romanian women with breast and ovarian cancer. They have evaluated 250 breast cancer and 240 ovarian cancer patients for the pathogenic germline mutations using NGS technology. 

The study have identified 35 BRCA gene mutations among breast cancer and ovarian cancer patients, 18 variations in the BRCA 1 and 17 variations in the BRCA2 gene. The sequencing have uncovered common BRAC1 and BRAC2 mutations among the Romanian population. They have also identified that the most prevailing BRAC1 mutations c.5266dupC and c.3067C>T found to have associated with triple negative breast cancer. These mutations were also highly prevalent in the OC patients. They have also identified the most common BRAC2 mutation c.9371A>T, which was also reported earlier in the Romanian population, observed both in the BC and OC patients in this study. Another BRAC2 mutation c.9371A>T, which was also reported earlier in the Romanian population found to have associated with early onset of BC in this study.

In addition, BRAC2 mutation c.4022C>G, which was not previously reported with BIC was reported first time associated with BC in this study. 

Altogether this study systematically unravel the pathogenic mutations associated with breast and ovarian cancer in the Romanian population, which may provide valuable insight identifying individuals at higher risk for hereditary breast and ovarian cancers in this population. 

Minor Comments:

1. Please change the decimal points to comma in the mutation numbers provided in the introduction part (lane 58-59).

Reviewer 2 Report

In this study, Vidra et al have evaluated breast cancer and ovarian cancer patients of Romanian ethnicity for the germline mutations in the risk genes. They identified several mutations in the BRCA1 and BRCA2 gene; c.3607C>T (BRCA1) and c.9371A>T (BRCA2) being the most prevalent. These are important findings in terms of assessing the risk of breast and ovarian cancers in the Romanian population.

The authors need to address the following concerns:

Major:

1. The study reveals several mutations in the BC and OC patients. Are there any specific mutations associated with enhanced relapse? Do the common treatment regimens have an impact on these mutational signatures?

2. What is the prevalence of tumorigenesis with the new mutation that was identified in this study? How common is this mutation in the general Romanian population?

3. The discussion section seems to be a mere reiteration of results. It needs to be more scholarly. Some topics that need to be covered include:

a. Significance of BRCA genes in BC and OC and how their findings align with the existing studies and what is new.

b. Interpretations and implications of their findings and how can they be exploited for diagnosis and treatment of these tumors.

c. How their findings from the Romanian population are similar or different from global statistics?

Minor:

1. The authors need to define every abbreviation for genes, proteins, and other words before using the short form in the entire text. For example, BRCA (Breast Cancer gene).

2. The title needs a slight edit: “Spectrum of BRCA 1/2 mutations in Romanian high-risk breast and ovarian cancer patients”. The high risk doesn’t fit well if you are directly studying the patients. The high risk will be valid if you are studying a first-degree relative or someone who shows genetic mutations but has not yet been diagnosed with cancer.

3. Regarding the representation of statistics, instead of stating “About 9.629 women are diagnosed with breast cancer and 1.840 women are diagnosed with ovarian cancer every year in Romania (Line 21)” – it is better to say “About 10 women are diagnosed with breast cancer and 2 women are diagnosed with ovarian cancer every year in Romania”. This rounding to the nearest whole number needs to be done for all such statistical references.

4. The manuscript needs extensive professional proofreading for correcting grammatical errors including punctuations, tenses, spaces, and spelling errors in the text as well as tables.

Reviewer 3 Report

Please check the attached file

Reviewer 4 Report

In this study, the authors described the spectrum of BRCA1/2 mutations in Romanian high-risk breast and ovarian cancer patients, evaluating the presence of damaging germline mutations in BRCA1/BRCA2 genes using NGS technology.

Main points:

- In the Introduction section, the authors could better describe the breast and ovarian cancer characteristics associated to the presence of damaging germline mutations in BRCA1/BRCA2 genes and reporting the most recent literature.

- In methods section, the authors could better clarify the criteria for high-risk breast and ovarian cancer patients selected for this study.

Moreover, the authors could divide three parts described in this section including:

  1. the description of the series of breast and ovarian cancer patients analyzed, including the part of informed consent for genetic testing (lanes from 95 to 98) and the clinical and pathological informations collected in this study;
  2. the description of genetic analysis, including the description of Next Generation Sequencing technology used (starting from the kit for library preparation, quality control of samples and reads sequenced and analyzed, sequencing platform used in the study, the tool used for variant annotation and classification….).
  3. the descriptive statistics, including the description of statistical tests used in the study.

- In the results section, the authors first describe the clinical and pathological characteristics for 35 Breast cancer patients and 60 Ovarian cancer patients with BRCA mutations, this information (number of mutated cases in BRCA genes) could be added in the title of 3.1 paragraph. Subsequently, the authors can describe the pathogenic and likely pathogenic mutation identified in two groups of patients, and their associations with the clinical and pathological characteristics in breast and ovarian cancer patients.

Round 2

Reviewer 2 Report

Most of the comments addressed.

Reviewer 3 Report

Please check the attached file 

Reviewer 4 Report

In materials and methods section revised,

the authors could shift the databases applied for clinical data interpretation in this part and not before (from line 161 to 189).

The authors described the MLPA and Sanger sequencing methods and the results of these applications are missing in the appropriate section. 
